# Latest Advances in Targeting the Tumor Microenvironment for Tumor Suppression

**DOI:** 10.3390/ijms20194719

**Published:** 2019-09-23

**Authors:** Chloé Laplagne, Marcin Domagala, Augustin Le Naour, Christophe Quemerais, Dimitri Hamel, Jean-Jacques Fournié, Bettina Couderc, Corinne Bousquet, Audrey Ferrand, Mary Poupot

**Affiliations:** 1Centre de Recherches en Cancérologie de Toulouse, Inserm UMR1037, 31037 Toulouse, France; chloe.laplagne@inserm.fr (C.L.); marcin.domagala@inserm.fr (M.D.); augustin.lenaour@inserm.fr (A.L.N.); christophe.quemerais@inserm.fr (C.Q.); jean-jacques.fournie@inserm.fr (J.-J.F.); bettina.couderc@inserm.fr (B.C.); corinne.bousquet@inserm.fr (C.B.); 2Université Toulouse III Paul-Sabatier, 31400 Toulouse, France; dimitri.hamel@inserm.fr (D.H.); audrey.ferrand@inserm.fr (A.F.); 3ERL 5294 CNRS, 31037 Toulouse, France; 4Institut Claudius Regaud, IUCT-Oncopole, 31000 Toulouse, France; 5Institut de Recherche en Santé Digestive, Inserm U1220, INRA, ENVT, 31024 Toulouse, France

**Keywords:** tumor microenvironment, resistance, TAMs, CAFs, T-regs, MSCs

## Abstract

The tumor bulk is composed of a highly heterogeneous population of cancer cells, as well as a large variety of resident and infiltrating host cells, extracellular matrix proteins, and secreted proteins, collectively known as the tumor microenvironment (TME). The TME is essential for driving tumor development by promoting cancer cell survival, migration, metastasis, chemoresistance, and the ability to evade the immune system responses. Therapeutically targeting tumor-associated macrophages (TAMs), cancer-associated fibroblasts (CAFs), regulatory T-cells (T-regs), and mesenchymal stromal/stem cells (MSCs) is likely to have an impact in cancer treatment. In this review, we focus on describing the normal physiological functions of each of these cell types and their behavior in the cancer setting. Relying on the specific surface markers and secreted molecules in this context, we review the potential targeting of these cells inducing their depletion, reprogramming, or differentiation, or inhibiting their pro-tumor functions or recruitment. Different approaches were developed for this targeting, namely, immunotherapies, vaccines, small interfering RNA, or small molecules.

## 1. Introduction

Resistance to cancer treatment is not only related to intrinsic properties of tumor cells but also to other cellular or acellular parameters of the tumor microenvironment (TME) [1]. In general, the TME is a highly heterogeneous and dynamic network, composed of normal and cancerous tissue-resident cells, a large proportion of recruited immune cells, and extracellular matrix components. After the initial state of the tumor, the TME (in particular, the immune TME) is modified to support and promote the tumor development while suppressing immune cell-mediated cytotoxicity [2]. The cellular composition of the TME is highly variable and could include almost all immune cell types such as CD8, CD4, and γδ T-lymphocytes, macrophages, natural killer (NK) cells, dendritic cells (DC), B-cells, and mast cells, as well as fibroblasts, neuroendocrine cells, adipose cells, endothelial cells, and mesenchymal cells [3]. All these cells, including tumor cells, take part in the tumor progression by promoting tumor growth, tumor dormancy, tumor invasion, and metastasis [4,5]. Tumor cells are able to release pro-tumor factors directly or indirectly by inducing hypoxia or necrosis, and by modifying other actors of the TME in its favor. Tumor-infiltrating immune cells (T-cells and macrophages), fibroblasts, and mesenchymal cells drive tumor progression. Depending on the tumor type, infiltrated immune cells can be found in variable proportions and display heterogeneous phenotypes with either pro- or anti-inflammatory properties [6]. In the latter case, despite their dual role in cancer, regulatory T-cells (T-regs) are a subset of immunosuppressive infiltrated T-cells mostly involved in the immune escape [7]. Tumor-associated macrophages (TAMs) or cancer-associated fibroblasts (CAFs) can also produce a favorable environment for cancer cells, as well as mesenchymal stromal/stem cells (MSCs), as shown in Figure 1.

These cells are considered as targets in cancer in order to improve classical therapies or to find new strategies to reverse treatment resistance. In this review, we focus on the latest advances in targeting of TAMs, T-regs, MSCs, and CAFs which could provide new therapeutic approaches in cancer. 

## 2. TAMs

### 2.1. Overview on Normal Macrophages and Their Physiological Functions

In normal tissue, resident macrophages have site-specific phenotypes, for example, microglia in the brain, Kupffer cells in the liver, alveolar macrophages in the lung, and peritoneal macrophages in the gut. These tissue-resident macrophages likely arise from embryonic precursors and are maintained by self-renewal [8]. However, all these macrophages display a common and principal role in maintaining steady-state homeostasis [9]. On the other hand, monocytes from the blood are recruited and transformed into mature macrophages in specific tissues, in response to cytokines and chemokines released during inflammation. Depending on the microenvironment, macrophages are differentiated into either classical activated (M1/Th1-activated) macrophages or into alternatively activated (M2/Th2-activated) macrophages [10]. 

M1 macrophages display pro-inflammatory properties involved in the resolution of infections and exert anti-tumor activities. Indeed, macrophages recognize pathogen-associated molecular patterns (PAMPs) such as lipopolysaccharides through Toll-like receptors (TLRs) at their surface, activating transcription factors (e.g., interferon regulatory factors and nuclear factor kappa B) and initiating the inflammatory response [11]. These M1-oriented macrophages, which could also be induced by cytokines secreted by other Th1 cells (e.g., IFN-γ), release pro-inflammatory cytokines (e.g., IL-1β, IL-6, IL-12, IL-23, and TNF-α) and type-1 cell-attracting chemokines (e.g., CXCL9 and CXCL10), favoring the recruitment of more macrophages and leucocytes to eliminate pathogens, characterized by the production of cytotoxic nitric oxide (NO) derived from the metabolism of arginine by inducible nitric oxide synthase (iNOS) [12]. There are no characteristic receptors at the surface of M1 macrophages, but they express CD16, CD86, CD80, IL-1RI, major histocompatibility complex class II (MHC-II), TLR2, and TLR4. 

On the contrary, M2-oriented macrophages are involved in the wound-healing process and allergies, and they display pro-tumor activities [10,13]. M2 polarization is induced by the release of IL-4, IL-13, or IL-10 by damaged cells, resulting in activation of the STAT6 transcription factor through the JAK–STAT pathway [14]. M2 macrophages are characterized by many specific receptors, e.g., IL-1R II, the scavenging receptor I (CD204), the mannose receptor (CD206), and the hemoglobin scavenger receptor (CD163) [15]. These cells produce large amounts of anti-inflammatory cytokines such as IL-10 and chemokines such as CCL17, CCL22 and CCL24 [12,16]. However, all these macrophages are never in a completely blocked phenotype due to their plasticity and ability to switch between phenotypes depending on microenvironment cues. Wound sites can also be sensitive to pathogen attacks, and both macrophage types exist within this type of environment, essential for wound resolution [17]. 

### 2.2. Tumor-Associated Macrophages

Within tumors, macrophages can be in between or off the spectrum of the M1/M2 states depending on the tumor stage [18]. These TAMs can be derived from resident macrophages [19] or attracted from bone marrow and spleen to the tumor site thanks to the CCL2 (MCP-1) and CCL5 (RANTES) chemokines produced by the tumors, fibroblasts, endothelial cells, and even by the TAMs themselves [20]. Colony-stimulating factor-1 (CSF-1) is also a monocyte/macrophage attractant, as well as a macrophage survival and polarization signal, which is highly important in tumors [21]. Early in tumor development, M1 macrophages are able to kill and remove tumor cells, and to produce cytokines for the recruitment and activation of other immune effector cells [22]. Nevertheless, the evolution of the TME, depending on the stage of the tumor and on the type of cancer, can revert this anti-tumor program and favor a switch of infiltrated macrophages into an M2 phenotype with pro-tumor and immune-suppressive functions [23]. Pro-tumor functions of TAMs are the result of a macrophage polarization with a particular cytokine profile (macrophage colony-stimulating factor (M-CSF), IL-4, IL-13, IL-10, Prostaglandin E2 (PGE2)) of the TME [24]. However, chemokines and signals in the TME originating from tumor cells, B-cells, and stromal cells (e.g., CSFs, TGF-β, IL-1) can deviate macrophage functions and phenotype, astray from the classical M1/M2 polarized cells. However, in general, TAMs stimulate tumor cell proliferation, migration, and genetic instability, and can protect them from apoptosis or therapy. TAMs are, thus, able to promote tumor development at primary sites, invasion, metastasis, angiogenesis, and lymphangiogenesis [25]. These cells express at their surface the scavenger receptor, CD163, and the mannose receptor, CD206, and also have immunosuppressive functions neutralizing recruitment and functions of cytotoxic CD8 T-cells and natural killer cells through the secretion of IL-10 and TGF-β in the TME [26,27]. TAMs also represent a strong source of iron, which is essential to tumor development [28]. Interestingly, several meta-analyses of more than 55 studies of different cancers showed that high infiltration of TAMs correlated with poor overall survival in the majority of cancers, indicating that infiltrating TAMs preferentially display a pro-tumor phenotype [29]. Through their pro-tumor functions, TAMs are also able to promote the recovery of tumors in mouse models following chemotherapies, radiotherapies, or immunotherapies [30].

For all these reasons, TAMs are good candidates to target in TME, and several therapy strategies modulating TAM functions, infiltration, and number are emerging. Figure 2 summarizes the different drugs targeting TAMs addressed in this review.

### 2.3. TAM Targeting

#### 2.3.1. Differentiation and Depletion

TAMs can be depleted by blocking CSF-1 receptor (CSF-1R), a tyrosine kinase receptor whose binding with CSF-1 or IL-34 leads to a signaling cascade promoting proliferation, function, and survival of macrophages [31]. Several antagonists and antibodies to CSF-1R were developed for the therapeutic modulation of infiltrating TAMs and tested in preclinical models. In a preclinical breast cancer model, anti-mouse CSF-1R antibodies were shown to deplete macrophages and to reduce tumor size [32]. Treatment with RG7155l, a humanized monoclonal antibody, was associated with a reduction in tumor-infiltrating TAMs and an increase in infiltration of effector T-cells in a fibrosarcoma mouse model, as well as in colorectal cancer and cancer patients with advanced tenosynovial giant cell tumors [31].

Alongside antibodies blocking CSF-1R, small-molecule antagonists to CSF-1R, capable of penetrating the blood–brain barrier, were been developed. For example, PLX3397 (pexidartinib) was associated with a reduction in the number of tumor-associated microglia and glioblastoma invasions in glioma mouse models [33]. In an ovarian cancer model, GW2580 induced a reduction of ascite volume and a decrease in the number of infiltrating macrophages [34]. In a phase II clinical trial, patients with recurrent glioblastoma displayed a good tolerance to GW2580, but only 8% of 37 patients showed a six-month progression-free survival [35]. 

To enhance the potential of these inhibitors, combination strategies with conventional chemotherapy were proposed. The efficacy of paclitaxel in a mammary adenocarcinoma mouse model was enhanced by the blockade of CSF-1R/CSF-1, with paclitaxel upregulating the production of CSF-1 [36]. However, a recent phase I study using a combination of a monoclonal anti-CSF-1R with paclitaxel in patients with advanced and metastatic solid tumors revealed TAM depletion but no clinically relevant anti-tumor activity [37]. In the case of gemcitabine, its activity was enhanced by its combination with GW2580 in the treatment of mice with pancreatic adenocarcinoma, through the upregulation of cytidine deaminase causing resistance to gemcitabine [38]. More recently, combination of anti-CSF-1R with docetaxel (microtubule-stabilizing agent) revealed TAM depletion in a murine epithelial ovarian cancer model, in correlation with the anti-tumor effect of docetaxel [39]. Moreover, PLX3397 was able to potentiate the therapeutic effects of radiotherapy in a preclinical mouse xenograft model with intracranial human glioblastoma through the neutralization of CSF-1, which was upregulated by radiotherapy [40]. 

Finally, it was shown that CSF-1R targeting by the BLZ945 inhibitor could lead to the switch of the TAM phenotype toward an anti-tumor phenotype in a glioma preclinical model thanks to GM-CSF and IFN-γ produced by glioma cells concomitantly to TAM depletion [21].

Another way to deplete TAMs is to use bisphosphonates (BP), which induce apoptosis of myeloid cells by inhibiting the essential farnesyl diphosphate synthase, a key enzyme for cholesterol synthesis and protein prenylation. These compounds have a high affinity for bone hydroxyapatite, where they are internalized by bone osteoclasts leading to their apoptosis, and they are, thus, indicated as treatment in pathologies with abnormal bone resorption such as osteoporosis or multiple myeloma [41]. However, all tissue-resident macrophages, including TAMs, can be targeted by BPs such as clodronate in a liposomal formulation with an increase in survival in some preclinical cancer models correlated with macrophage depletion [42]. In a murine xenograft model for cutaneous T-cell lymphoma, the reduction of tumor growth was associated with a decrease in pSTAT3 and macrophage number [43]; the same association was observed in other preclinical mouse models of bone metastasis from lung cancer [44], lung metastasis from breast cancer [45], or melanoma xenograft models [46]. At the clinical level, clodronate was shown to reduce the incidence of bone and visceral metastasis in human mammary carcinomas treated with a combination of clodronate/chemo- or hormonal therapy [47], and, in patients with prostate cancer, another BP reduced skeletal-related events and improved progression-free survival time [48].

Finally, trabectedin, a tetrahydroisoquinoline alkaloid with anti-neoplastic effects, was used for the treatment of advanced-stage tissue sarcoma and relapsed ovarian cancer and was shown to cause a partial depletion of circulating monocytes and TAMs [49]. Trabectedin activates a TRAIL-dependent pathway of apoptosis, and myeloid cells expressing very low levels of TRAIL are insensitive to TRAIL triggering of apoptosis [50].

#### 2.3.2. Reprogramming 

Even if TAMs generally display a pro-tumor role, they can be converted into anti-tumor cells and activate the immune system depending on the context. Regarding this plasticity, TAMs can be reprogrammed toward a tumoricidal phenotype to restore their anti-tumor properties thanks to the targeting of surface markers or signaling molecules, or to modifications in the metabolism.

Several markers at the surface of TAMs can be targeted to switch their phenotype such as the scavenger receptor MARCO, TLRs, CD40, or CCR5 [51,52,53,54]. The pattern recognition scavenger receptor MARCO was linked to clinical outcome [51]. An anti-MARCO monoclonal antibody was, thus, developed and was shown as having an anti-tumor activity in breast and colon carcinoma, and in melanoma models through reprogramming of TAMs populations to a pro-inflammatory phenotype and increasing tumor immunogenicity. This was shown as being dependent on the inhibitory Fc-receptor, FcγRIIB [51]. The pro-inflammatory phenotype can also be activated by specific ligands of TLRs or of CD40. Some TLR synthetic ligands were tested in cancer, such as the TLR3 agonist Poly(I:C) which activates the NF-κB pathway, leading to pro-inflammatory M1 polarization with production of type I IFN [55]. Intratumoral delivery of a TLR9 agonist, such as cytosine–phosphate–guanine oligodeoxynucleotides (CpG ODN), or a TLR7/8 agonist (R848) showed a tumoricidal activity in mouse models of melanoma and breast cancer [52,56]. Clinical trials with IMQ and a DNA-based TLR9 immunomodulator in metastatic cancer also showed histological tumor regression and an increase in lymphoid immune infiltration [57]. TLRs can also be activated by sensing bacterial ligands such as the attenuated ΔactA/ΔinlB strain of *Listeria monocytogenes*, which, when introduced into the aggressive ID8-Defb29/Vegf-A murine ovarian carcinoma, is preferentially phagocytosed by TAMs and reprograms the population from one of suppression to immunostimulation [58].

CD40 is also an important receptor at the TAM surface, as its ligation with the CD40 ligand at the T-cell surface stimulates T-cell–based anti-tumor responses. In preclinical studies, a rat anti-mouse CD40 antibody showed remarkable therapeutic activity in the treatment of CD40^+^ B-cell lymphomas [59]. The first class of modified anti-CD40 antibody, presenting five point mutations in the Fc domain (CP-870,893), showed efficient immunostimulation of effector T-cells in mouse models but little clinical effect in advanced cancer patients [60]. Recently, the same CD40 agonist was shown to increase pancreatic carcinoma sensitivity to chemotherapy in a PDAC mouse model associated with a switch of infiltrated macrophages due to an increase in CCL2 expression and IFNγ in these mice [54]. This last study opened promising perspectives for combination therapies. CCR5 antagonists such as maraviroc provided an anti-tumor effect in a phase I trial in patients with liver metastasis of advanced refractory colorectal cancer correlated with a macrophage repolarization [53]. 

Furthermore, a majority of cancer cells express a high amount of CD47 at their surface, which interacts with signal regulatory protein alpha SIRPα on myeloid cells such as macrophages to transmit the “do not eat me” signal. This inhibition of phagocytosis of autologous cells exists in homeostatic conditions by the prevention of myosin IIA accumulation at the phagocytic synapse. Anti-CD47 antibodies or engineered SIRPα–Fc fusion were shown to restore macrophage ability to phagocytose cancer cells and prime cytotoxic CD8 T-cells, leading to the tumor regression in several tumor models and in clinical trials. Two anti-CD47 antibodies (Hu5F9-G4 and CC-90002) and one soluble recombinant SIRPα-crystallizable fragment (Fc) fusion protein (TTI-621) are currently being tested in phase I clinical trials in AML and other hematological malignancies, pediatric brain tumors, and some multiple solid tumors (trial numbers NCT02678338 and NCT03957096). To minimize the off-target toxicity (transient anemia), bispecific antibodies targeting both CD47 and tumor-associated antigens were recently developed and obtained promising results [61]. 

Specific pathways involving STAT3 or STAT6 characteristic of an anti-inflammatory response can be targeted to modify M2 characteristics of TAMs. For instance, through its ligation to the IL-10 receptor, IL-10 and IL-6 induce the activation of STAT3 necessary to drive an anti-inflammatory effect. Formerly, microtubule-stabilizing agents such as paclitaxel were shown to promote polarization of MDSC to macrophages with the M1 phenotype via suppression of STAT3 phosphorylation [62]. The STAT3 axis was then more specifically targeted with an intravenous injection of a STAT3–small interfering RNA (siRNA)–CpG conjugate which was shown to silence the immune suppressor gene *STAT3*, reprogramming the functions of TAMs correlated with an immune-mediated tumor rejection [63]. A recent in vitro study showed that corosolic acid, packaged within long-circulating liposomes and coupled to an anti-CD163 antibody, inhibited the activation of STAT3 in human M2 type macrophages and their IL-10-induced gene expression [16]. In the same way, STAT6, one of the major signal transducers activated by IL-13, involved in M2 polarization, can be inhibited by different synthetic molecules (AS1517499, TMC-264, A771726), leading to an inhibition of tumor growth in the 4T1 mammary tumor model and modification of genetic markers for TAM infiltration [64]. 

Furthermore, microRNA (miRNA)-155 is best characterized as a pro-inflammatory miRNA since it enhances M1-like macrophage activation by decreasing inhibitors of pro-inflammatory responses and targets the IL-13 receptor to reduce alternative macrophage activation [65]. Currently, several miRNAs are implicated in the regulation of macrophage activation and polarization in cancer [66], and developments of pharmacological formulations that either suppress or enhance the activity of selected miRNAs to reprogram the TAM phenotype are emerging. For instance, targeting of miRNA-processing enzyme DICER in macrophages with lysozyme-M-driven Cre prompts M1-like TAM reprogramming in lung carcinoma and MC38 tumors [67].

The switch of M2 TAMs toward an M1-like phenotype can also be driven by rapamycin, which is an mTORC1-specific inhibitor, inducing an anti-tumor effect in a hepatocarcinoma mouse model [68]. Moreover, inhibitors of PI3kγ, a molecule upstream of mTOR, are able to promote TAM-immunostimulatory responses in several cancer models [69]. Activation of MAPKs via production of ROS, thanks to a copper chelator, CuNG (*N*-(2-hydroxy acetophenone) glycinate), can convert suppressive TAMs toward a pro-inflammatory phenotype [70]. M1 genes such as *iNOS* and *CXCL9* can also be upregulated concomitantly with the downregulated expression of M2 genes, *arginase-1* and *CD206*, thanks to the inhibition of the CYP4X1 monooxygenase or the activation of the autophagy-induced RelB/p52 by flavonoids in glioma or hepatocellular carcinoma models [71,72]. Several synthetic molecules can also actively reverse the M2 phenotype of TAMs such as TMP195, an HDAC inhibitor, in breast cancer mouse models, or paclitaxel, an anti-microtubule agent, in patients with ovarian carcinoma [73,74].

Combination strategies were also reported to have a good effect on TAMs reprogramming. Anti-CD40 was combined with anti-CSF-1R to create a pro-inflammatory environment eliciting T-cell responses before depletion of TAMs [75] or with chemotherapies such as imatinib (tyrosine kinase inhibitor) with strong tumor regression in a gastrointestinal tumor mouse model [76]. A TLR9 agonist (IMO-2055) combined with chemotherapies and anti-VEGF showed anti-tumor activity in advanced or metastatic lung cancer [77]. Moreover, TLR9 agonists were also able to enhance radiofrequency-induced CTL responses, potentiating the inhibition of primary tumor growth and lung metastasis [78]. 

Regulating macrophage mitochondrial function is another way to activate the reprogramming of TAMs. Indeed, downregulation of the gene encoding pyruvate dehydrogenase is avoided by dampening the nuclear respiratory factor 1 degradation under hypoxia, which minimizes the Warburg effect and promotes M1 polarization of TAMs [79]. Moreover, cancer cells secrete lots of lactate through glycolysis, which is recognized by Gpr132 at the surface of TAMs and promotes M2 polarization. PPARγ agonists, which are suppressive for the Gpr132 axis, or siRNA silencing of Gpr132, was successfully used to desensitize TAMs to lactate stimulation in a breast tumor model [80,81].

Finally, recent studies identified novel drug targets for further investigation in the future. For instance, loss of *Gadd45b*, a gene involved in the regulation of growth and apoptosis in myeloid cells, restored activation of pro-inflammatory functions of TAMs and intra-tumor immune infiltration in a fibrosarcoma mouse model [82]. Interfering with angiopoietin family molecules might be an effective strategy for reprogramming TAM polarization in NSCLC [83]. CaMKK2 was defined as a new myeloid-selective checkpoint and could be blocked to facilitate a favorable reprogramming of immune cells in the TME [84]. Several studies also proposed to reprogram the energetic metabolism in TAMs, in which fatty-acid oxidation is preponderant, decreasing lipid and cholesterol intake and metabolism [85,86]. 

#### 2.3.3. Inhibition of Recruitment

Modulation of the recruitment of monocytes and macrophages to the tumor site in order to disrupt TAM populations represents a promising complementary therapeutic strategy. Inhibition of CCR2/CCL2- and CSF-1R-dependent signaling pathways is a major focus in this field, as direct targeting of CCR2/CCL2 showed positive outcomes in various experimental cancer models [87,88,89,90,91]. However, clinical translation proved to be problematic. Clinical trials with agents depleting CCL2 (carlumab or CNTO888) or blocking CCR2 (MLN1202) showed poor therapeutic responses when used as monotherapies [92,93]. In fact, administration of an anti-CCL2 antibody initially showed decreased levels of CCL2 but led to elevated levels of this chemokine in the long term. Moreover, additional precaution must be taken, as discontinuation of CCL2 therapy in murine models of breast cancer showed increased monocyte mobilization and tumor progression [92]. Phase I and II clinical trials with carlumab were carried out in patients with different type of tumors and indicated that CCL2 levels were only partially suppressed associated with no therapeutic efficacy [94,95]. The blockade of CCR2 by PF-04136309 was assessed in a phase Ib trial in patients with pancreatic cancer in combination with folfirinox chemotherapy. Compared to no response with the monotherapy, 97% of patients with the combination showed an objective tumor response and others showed a local tumor response [96]. These results showed high complexity and mechanistic redundancy within the CCR2/CCL2 axis, which still needs to be addressed. An alternative approach could be the indirect inhibition of the CCR2/CCL2 pathway. It was observed that, in a mouse model of glioblastoma, TAMs express aryl hydrocarbon receptor (AHR), which, upon activation by tumor released kynurenine (KYN), upregulates CCR2 expression. Blocking the AHR led to decreased TAM infiltration and tumor growth with concomitant increased survival of the animals [26].

Targeting of CSF-1R also showed a decrease in macrophage/monocyte recruitment [97]. In spite of this, numerous clinical trials showed a limited therapeutic effect when CSF-1R was targeted alone [98], with the exception of benign tenosynovial giant cell tumors, where a decrease in TAM infiltration and disease stabilization was achieved [99]. Efficacy of CSF-1R inhibitors is still being tested in anticancer therapies, also in the form of broad-spectrum kinase inhibitors such as chiauranib (CSF-1R, VEGF1-3, PDGFRα, c-Kit, Aurora B inhibitor) [100] and pexidartinib (CSF-1R, KIT, FTL3 inhibitor), with potential anticancer results [101]. 

Research on a mouse model of lung adenocarcinoma showed that FGF released by CAF induced TAM recruitment. Blocking of FGF receptor (FGF-R) with the Fgfr tyrosine kinase family inhibitor AZD4547 showed strong TAM abolishment and tumor regression, making this receptor a potential therapeutic target [102]. In addition, targeting other important molecules for monocyte/macrophage recruitment in cancer, i.e., catecholamines by 6-hydroxydopamine [103] and complement components such as C5a, may also be promising in cancer treatment [104]. 

## 3. T-Regs 

### 3.1. Overview on T-Regs and Their Physiological Functions

T-regs are a lymphoid cell population with a key function in modulating an immune-suppressed TME. T-regs are a subset of CD4^+^ T-cells identified by expression of transcription factor forkhead box P3 (FOXP3) and can be defined as CD4^+^ CD25^+^ FOXP3^+^ T-cells [105]. Compared to naïve T-regs, activated T-regs express higher levels of CC chemokine receptor 4 (CCR4) [106], CD28 [107], glucocorticoid-induced TNFR family related gene (GITR) [108], cytotoxic T-lymphocyte antigen-4 (CTLA-4) [109], lymphocyte-activation gene-3 (LAG-3, also known as CD223) [110], neuropilin 1 (Nrp-1) [111], programmed cell death-1 (PD-1) [112], T-cell immunoglobulin and mucin-domain containing-3 (TIM-3) [113], T-cell immunoreceptor with Ig and ITIM domains (TIGIT) [114], and V-domain Ig suppressor of T-cell activation (VISTA) [115]. T-regs are essential to maintain immunity equilibrium (i.e., self-tolerance) and resolve inflammation [116]; depletion of T-regs leads to autoimmune diseases in murine models [117]. T-regs regulate immune responses through four major mechanisms [118]: (i) secretion of inhibitory cytokines such as IL-10, IL-35, and TGF-β; (ii) direct cytolysis of target cells by the secretion granzyme-A, granzyme-B, and perforin [119]; (iii) IL-2 deprivation due to a high-affinity for CD25 (IL-2 receptor) [120]; and (iv) modulation of maturation and/or function of DC which permits activation of effector T-cells. Moreover, it was shown that LAG-3, a high-affinity binder of the MHC class II molecule required for maximal T-reg suppression, could abort DC maturation through induction of an immune receptor tyrosine-based activation motif (ITAM)-mediated inhibitory signaling pathway [110]. 

### 3.2. Tumor-Infiltrating T-Regs

T-regs play a pernicious role in a tumor context by suppressing anti-tumor responses which limit the efficacy of cancer immunotherapy [111,121]. T-regs could infiltrate many solid human tumors since there are several studies which report an increased proportion of T-regs in the TME of melanoma patients [121], as well as lymphoma [122], breast and pancreatic cancers [123], and ovarian and lung [124] cancers. An increase in the proportion of tumor-infiltrating T-regs (TI T-regs) is often linked to a poor survival rate in many cancers. Despite the development of immunotherapies showing spectacular clinical responses, some patients do not respond to these therapies. As T-regs are mainly present in tumor tissues, they constitute an attractive target in T-cell infiltrating tumors. Therefore, depleting effector T-regs would modify the equilibrium in TME from immune suppression to active anti-tumor activity, while preserving a continual supply of effector T-regs from intact naïve T-regs.

The latest therapeutic advances targeting T-regs by using antibodies, vaccines, or small molecules specific for T-reg features are summarized in Figure 3.

### 3.3. T-Reg Targeting

#### 3.3.1. Antibodies Depleting Tumor-Infiltrating T-Regs 

Several antibodies were found to deplete T-reg cells including basiliximab, a chimeric anti-CD25 monoclonal antibody, which was first tested in kidney transplant patients and showed a transitional loss of FOXP3^+^ and FOXP3^−^ CD25^+^ T-cells in the circulation [125]. Daclizumab, an FDA-approved monoclonal antibody against the CD25 receptor, showed a durable decrease of circulating T-regs without impairing vaccine-induced T-cell responses, reducing CD25 and FOXP3 expression of T-regs and increasing their secretion of IFN-γ [126]. 

The anti-CCR4 antibody displayed an effective role in specifically depleting effector T-regs and increasing the induction of tumor antigen-specific CD4 and CD8 T-cells in vivo [106]. Indeed, antibodies against CCR4 decrease the proportion of TI T-regs in murine tumors [127], and a humanized anti-CCR4 antibody, KW-0761, is being tested in early clinical trials in patients [128]. Recent studies showed that targeting the CCL22–CCR4 axis could also deplete T-regs and block their interaction with DC, allowing a stronger immune response as DCs regulate adaptive immunity through the constitutive expression of CCL22 [129]. Interestingly, treating T-cells with the CCR4 ligand, CCL17, can disable the production of IFN-γ, thus suggesting chemokines as factors that can directly reprogram the functionality of T-cells [127]. 

An agonistic anti-GITR antibody (non-depleting) can reduce the T-reg cell-mediated suppressive function and increase the effector function of conventional T-cells in mouse models [108]. The agonistic antibody for GITR is already being tested on melanomas and other advanced solid tumor patients. Recently, this antibody was shown to lead to an important reduction of TI T-regs in mice [130] and could overcome resistance to immunotherapy in human solid cancers [131]. This study showed a decrease of peripheral and TI T-regs after anti-GITR antibody TRX518 (NCT01239134) treatment, providing an easily assessable biomarker of anti-GITR activity. Nonetheless, these results do not reveal a major clinical response despite T-reg depletion and increased T-effector:T-reg ratios in patients and in advanced tumor-bearing mice. Yet, combination with PD-1 blockade could overcome resistance of advanced tumors to anti-GITR monotherapy. Furthermore, antibodies against GITR, while capable of reprogramming T-regs, can also enhance the proliferation and cytokine production of intra-tumor CD8 T-cells [132]. 

CD28 is a co-stimulatory receptor that represents the strongest secondary stimulus for T-reg functionality. Indeed, blocking CD28 signaling using anti-CD28 antibodies completely inhibits their stability and function. Furthermore, using anti-CD28 in cancer impairs TI T-reg differentiation and function, reducing their capacity to suppress anti-tumor immune responses and promoting tumor control [107]. CD28 binding is opposed by CTLA-4, an inhibitory receptor that competes for binding to the same ligands, B7-1 and B7-2, on antigen-presenting cells [133]. 

Immune checkpoint blockade antibodies, such as anti-CTLA-4, anti-PD-1, or anti-TIGIT, were tested to deplete T-regs since they are upregulated both on CD8 T-cells and T-regs in tumors, enhancing anti-tumor T-cell functions while simultaneously disabling T-regs to potently enhance the liver cancer immune response [134]. Using an experimental model of Fc-receptor deficient mice, the anti-tumor effect of the anti-CTLA4 antibody was shown to be dependent on antibody-dependent cellular cytotoxicity of TI T-regs instead of re-activating conventional T-cells [135]. 

In cancer patients, strong correlations were reported between the clinical efficacy of ipilimumab and decrease in TI T-reg proportion. Some T-regs express PD-1, but possible effects of PD-1 blockade on PD-1-expressing T-regs in tumor tissues remain unknown [136]. A recent study showed that deletion or blockade of PD-1 increases proliferation and immunosuppressive activity of PD-1^+^ T-regs in humans and mice [137]. Apparently, PD-1^+^ effector T-regs could display a major role in hyper-progressive disease (HPD) development (i.e., rapid progression of the cancer instead of its regression in certain cancer patients treated with anti-PD-1 antibody), indicating that deletion of T-regs in tumor tissues could be effective in treating HPD during anti-PD-1 mAb therapy. CD28 blocking antibody can also be used to reprogram T-regs influencing effector T-cell activation, acquisition of the glycolytic metabolism essential for effector functions, and the effectiveness of PD-1 checkpoint blockade treatment. In tumor-bearing mice, B7 co-stimulation is required for successful PD-1 therapy. Moreover patients treated by PD-1 therapy exhibited CD8 T-cells expressing CD28 [138]. 

LAG-3 expression on T-regs was also shown to be required for maximal suppressive activity, as the blockade abrogates T-reg function in in vitro proliferation assays [110]. Additionally, transfection of LAG-3 in non-T-reg CD4 T-cells resulted in the acquisition of a regulatory phenotype, with reduced proliferation of co-cultured responder T-cells. More recent studies showed that LAG-3 promotes T-reg differentiation, while LAG-3 blockade inhibits T-reg induction [139]. This study also showed that blockade or genetic deletion of LAG-3 maintained CD4 T-cells in a Th1 phenotype, with LAG-3 restricting IL-2 and STAT5 signaling, adjusting their capacity to be suppressed by T-regs. 

However, unlike PD-1+/hi Tregs, TIM-3^+^ TI T-regs displayed higher a suppressive capacity due to a stronger expression of CTLA-4 and CD39. Increased ingress of TIM-3^+^ CD4 T-cells or TIM-3^+^ T-regs is associated with a poor prognosis in patients with various malignancies including non-small-cell lung cancer [140]. 

Several teams showed that TIGIT^+^ T-regs display higher immunosuppressive activity than TIGIT^−^ T-regs [141]. Nonetheless, the effect of the anti-TIGIT antibody on established tumors remains to be determined, especially if it acts on T-regs, CD8 T-cells, or both. 

The VISTA monoclonal antibody increases the proportion of tumor-specific T-cells in the periphery, leading to ingress, spread, and effector function of TI T-cells within the TME. VISTA blockade changes the suppressive function of the TME by decreasing the proportion of MDSC, increasing the rate of activated dendritic cells within the TME, and reducing the appearance of tumor-specific T-regs [142]. Another study showed that VISTA monotherapy did not avoid recruitment of T-regs in the TME [143]. Indeed, VISTA blockade leads to transition of CD8 T-cells into functional effector T-cells, but is not sufficient to shrink tumor growth due to weak T-reg suppression in the TME. Nonetheless, combination of VISTA with CTLA-4 blocking antibodies efficiently inhibited T-reg recruitment and increased the ratios of both CD8 T-cells/T-regs and CD4 conventional T-cells/T-regs in the TME, which will be more effective than combined PD-1 and VISTA blockade for tumors in which T-reg-mediated immune regulation is dominant (such as head and neck squamous cell carcinoma). 

Denileukin difitox (Ontak^®^) is a recombinant fusion protein product of the diphtheria toxin and IL-2 that selectively binds to the IL-2 receptor of cells and inhibits protein synthesis after its internalization [144]. Recently, a second-generation diphtheria-toxin-based fusion protein associated with a better tolerance (i.e., fewer vascular leak adverse events) showed interesting results both alone and in association with PD-1 blockade [145]. 

Depleting Nrp-1 in TME-infiltrating T-regs in a melanoma mouse model increases their ability to produce IFN-γ, which reinforces CD8 T-cell responses within the TME in a paracrine fashion and promotes the reprogramming of other TI T-regs into IFN-γ-producing cells [111].

#### 3.3.2. Tumor-Infiltrating T-Reg Vaccine Approaches 

As ectopic expression of FOXP3 in conventional T-cells confers an immunosuppressive phenotype, a strategy for targeting FOXP3 was developed to counteract imperfect immune responses against tumor cells. A recent study showed delayed tumor growth, decreased production of IL-10, IL-2, and TGF-β, and increased survival of mice after inoculation of the FOXP3-silenced B16F10 melanoma cell line compared to mice injected with the wild-type cell line [146]. Moreover, a tumor cell vaccine associated with *FOXP3* gene silencing can improve the efficacy of therapeutic anti-tumor vaccination [147]. Furthermore, a recent study showed that vaccination of mice with Fox–Fc DNA vaccine/recombinant FOXP3–Fc fusion protein induced a CTL response against FOXP3^+^ T-regs, which decreased tumor growth and prolonged survival rates [148]. These results showed that the FOXP3 vaccine displays an immune response against tumors by targeting both T-regs and MDSC, which could be used as a potential immunotherapy approach [149]. 

#### 3.3.3. Small Molecules for T-Reg Depletion or Functional Modulation

Repeated exposition of high-dose chemotherapy, cyclophosphamide, an alkylating molecule which interferes with DNA replication, kills proliferating cells and impacts all T-cells. Low-dose administration of cyclophosphamide over a long period was shown to selectively deplete highly proliferating T-regs in tumor tissues, and enhance anti-tumor immune responses in humans and rodents [150,151]. Low doses of cyclophosphamide deplete TI T-regs in metastatic colorectal cancer patients [152]. Several studies combined chemotherapeutic agents such as cyclophosphamide with other drugs targeting T-regs [153]. 

TCR signaling molecules which are differentially controlled in T-regs in comparison with conventional T-cells can also be targeted. Indeed, ZAP-70, which is specifically repressed in T-regs upon TCR activation, can be targeted to abrogate TCR signaling by interfering with TCR proximal signaling molecules, resulting in selective death of T-regs, in particular effector T-regs [154].

Moreover, anti-tumor immune responses were increased by an inactivating mutation (D910A mutation) in phosphatidylinositol-3-kinase (PI3K) p110δ or a knockout of PI3K in T-regs in mice, without autoimmunity in the mutant mice [155]. However, PI3K activity seems to be essential for T-reg survival and function. Indeed, genetic deletion or pharmacological inhibition of the PI3K subunit p110δ selectively impairs TI T-reg function and favors anti-tumor immune responses [156].

T-regs regulate immune responses trough the secretion of inhibitory cytokines such as TGF-β, IL-10, and IL-35. Their increase in tumors is associated with a poor prognosis in various cancer types. TGF-β promotes the differentiation of induced T-regs in vitro [157]. Deletion of IL-10 in T-regs induces spontaneous colitis, highlighting the physiological importance of T-reg-derived IL-10 [158]. T-reg-derived IL-10 alters the myeloid compartment in the TME, indirectly providing regulation of T-cell-mediated anti-tumor immune responses through upregulation of T-cell stimulatory molecules such as major histocompatibility complex class II and CD80 on intra-tumor DCs [159].

Finally, the next challenge in T-reg targeting will be to use optimized antibodies specific for TI T-regs or engineered IL-2 molecules which do not bind T-regs [160]. Future generations of T-reg-based immunotherapies must consider (i) a suitable combination of targets to promote effector responses, (ii) abolishing specific TME T-reg infiltration or function, and (iii) determining the appropriate timeline of therapeutic administration leading to a better benefit/risk ratio.

## 4. MSCs

### 4.1. Overview on Normal MSCs and Their Physiological Functions 

Mesenchymal progenitor cells were firstly isolated three decades ago from bone marrow (BM-MSC). Since this first characterization, it was shown that MSCs can be isolated from most tissues including fat tissue (adipocyte-derived mesenchymal stem cells), skin, heart, kidney, etc., or from perivascular space (pericyte-derived MSCs) [161,162]. They are capable of differentiating into fibroblasts, adipocytes, osteoblasts, chondroblasts, vascular and perivascular structures, etc. They could be isolated on the basis of their ability to adhere to the plastic and for the expression of CD73, CD90, and CD105 markers. They do not express CD45, CD34, CD14, CD19, and human leucocyte antigen DR (HLA-DR) [161]. MSCs possess hallmark characteristics of stem cells or at least progenitor cells with regard to their self-renewal and differentiation properties [162]. MSCs could be used as therapeutic agents for regenerative medicine as they could contribute to tissue healing, mainly through the secretion of paracrine factors such as cellular adhesion molecules (including VCAM-1 and ICAM-1), growth factors (including TFG-β, EGF, HGF), cytokines (such as IL-1α, IL-1β, IL-6, and IL-8), angiogenic factors (such as VEGF and PDGF), and immunomodulatory molecules (PGE2). In addition, MSCs play a role in immune tolerance, through their immunosuppressive effects on T- and B-lymphocytes, and their influence on macrophage and dendritic cell polarization, thus preventing immunoreaction in both physiological and pathological conditions [163,164]. 

### 4.2. MSCs in Cancer

Although MSCs can halt cancer progression by inducing apoptosis, suppressing signaling pathways, initiating cell-cycle arrest, and increasing infiltration of monocytes and granulocytes [165,166], the majority of publications report pro-tumor activities. MSCs are able to stimulate tumor growth, increase angiogenesis, and favor metastasis development mainly through the release of activating factors such as cytokines and growth factors including IL-6 [167] and GDF15 in hematological malignancies [168]. 

In solid cancers, following recruitment to the tumor site, MSCs as CA-MSCs (cancer-associated MSCs) are distinct from the original cells. These infiltrated MSCs were described as having a role to play in ovarian cancer progression by releasing IL-6, CXCR1/2 ligands (CXCL1, CXCL2, and IL-8), and related cytokine leukemia inhibitory factor (LIF) [167,169]. MSCs are also able to promote tumor growth and metastasize through direct cell-to-cell contact or through their secreted exosomes, which are able to carry different molecules such as IL-6, TFG-β1, CCL2, and fibronectin, or miR-375, miR340 and miR-155 [162]. These exosomes also participate in tumor homing in vivo, indicating that MSCs play a role in metastasis [170]. The mechanisms via which exosomes promote cell proliferation, increase cell survival, activate the AKT and the MAPK pathways, and also inhibit p38, p53, and JNK pathways are being studied. 

They also were described to enhance vascular endothelial growth factor (VEGF) expression in tumor cells by activating the extracellular signal-regulated kinase 1/2 (ERK1/2) pathway [171]. Moreover, MSCs can inhibit the immune response mediated by NK cells [172], T-lymphocytes, or macrophages [173]. They are able to inhibit T-cell proliferation through the secretion of TGF-β and HGF, which induce T-cell proliferation arrest in the G1 phase and apoptosis of activated T-cells [172]. MSCs are also able to alter the activation and the differentiation process of T-cells and promote the generation of T-regs. In addition, MSCs could suppress the proliferation, differentiation, and activation of B-cells [172]. Through their inhibition of the T/B-cell immune response, MSCs are involved in tumor progression as tumor cells will not be recognized by the cellular and humoral effectors of the immune response. 

MSCs could also inhibit the activation of dendritic cells, downregulating their endocytotic and IL-12 secreting activities [172] and activating the generation of TAMs through the release of CXCR1/2 ligands [174]. Finally, MSCs are also implicated in the acquisition of chemoresistance via i) the release of soluble factors by MSCs (IL-6, IL-8, VEGF), ii) an exchange of membrane proteins between MSCs and tumor cells, iii) the release of exosomes, or iv) through the direct interaction within tumor cells [175].

### 4.3. MSC Targeting

#### 4.3.1. Inhibition of Recruitment 

MSCs are recruited to the tumor site from the bone marrow in response to soluble factors produced by tumor cells, such as IL-1, SDF-1, CCL5, and more recently the aspartic acid protease cathepsin D [176,177,178]. They can also migrate with the tumor cells from the primary tumor to another site and promote the growth of distant metastasis [179]. In the hypothesis that the use of antibodies targeting these chemoattractive molecules could be a way to limit MSC recruitment to the tumor sites, no results on in vivo models are reported in the literature.

#### 4.3.2. Reprogramming

CA-MSCs closely associated with both solid and hematologic malignancies were genetically and phenotypically analyzed [180]. As an example in myeloma patients, CA-MSCs which express alterative cytokines (IL-6, GDF15, IL-8) have an abnormal proliferative capacity, and present distinctive gene expression profiles [181]. In other hematological malignancies, it was shown that tumor cells activate the AKT/mTOR pathway in MSC and elicit VEGF and HIF-1 production. With regard to solid cancers, Rafii et al. [175] and more recently Mishra and Banergee isolated mesothelial-like stem cells (which presented a typical fibroblast-like phenotype) and identified the expression of several membrane markers that could be targeted such as CD9, CD10, CD29, CD73, CD146, and CD166 [182]. Finally, Vishnubalaji showed that MSCs could even transform into sarcoma cells and become neoplastic [180]. 

Taken together, all these observations led to the affirmation that CA-MSCs acquired different potencies when they were activated by the tumor microenvironment. They express several proteins that in theory could be targeted in order to eliminate CA-MSCs without affecting the other MSCs even if, to our knowledge, it is yet to be performed.

#### 4.3.3. Blocking Functions

Pro-tumor molecules such as IL-6, IL-8, and HGF secreted by CA-MSCs or their receptors can be blocked to circumvent the pro-tumor effects of the MSC. Pasquier et al. showed that tocilizumab (anti-IL-6R therapy) in association with chemotherapy significantly reduced ovarian cancer progression induced by MSC recruitment in an in vivo model [183].

Le Naour et al. showed that ovarian CA-MSCs overexpress CXCR1/2 ligands, which are involved in ovarian cancer progression and the acquisition of chemoresistance. They used a CXCR1/2 inhibitor to sensitize ovarian tumor cells to carboplatin and circumvent the pro-tumor effects of CA-MSCs. They showed that CXCR1/2 inhibition could be a potential therapeutic strategy to revert carboplatin resistance mediated by MSCs [184].

Another strategy could be to inhibit the effects of the MSC-released molecules. Castells et al. showed that MSCs secreted molecules able to activate the phosphatidylinositol 3-kinase (PI3K)/Akt signaling pathway, inducing the phosphorylation of the X-linked inhibitor of apoptosis protein (XIAP; caspase inhibitor from inhibitor of apoptosis protein (IAP) family). This IAP activation leads to a chemoresistance acquisition of the ovarian cancer cells (inhibition of apoptosis). Targeting XIAP in ovarian cancer could be of particular interest to circumvent pro-tumor effects of MSCs. SMAC mimetics such as DEBIO1143 could inhibit cIAP proteins. This compound was evaluated in both hematological or solid malignancies for its capacities to inhibit cIAP and promote the sensitization of tumor cells to chemotherapies both in mice models [185] and in clinical trials [186]. SMAC mimetics potentiate the effect of chemotherapy such as platinum salt. They induce tumor stabilization and regression. They are currently being studied in phase III clinical trials. 

#### 4.3.4. Blocking the Cell-to-Cell Contact

MSCs could act on tumor cells through a direct cell-to-cell contact. BM-MSCs may affect developing myeloma cell lines via direct cell-to-cell interactions. Breast and ovarian cancer cells at least transiently acquire new functional properties following interaction with MSCs via gap junctional intercellular communication or Notch signaling in vitro and in vivo. Several molecules and receptors are involved in the interactions between cells, such as ICAM-1 expressed on MSC and mucin-1 (MUC-1) on the cell surface of breast cancer cells [187]. Blocking Eph/Ephrin signaling affects cell attachment in vitro leading to cell death [188]. Cell-to cell contact and fusion are involved in chemoresistance acquisition mediated by MSCs, as this contact can induce membrane exchanges between the cells and the acquisition of new properties by the tumor cells (such as MDR proteins). Oncological trogocytosis and a fusion between the cells (hybrid) were described. The tetraspanin CD9 could be involved in the chemoresistance acquisition of the hybrid cells [175,189]. Targeting this molecule and inhibiting MSC/tumor cell fusion could be a way of inhibiting MSC pro-tumor activities.

#### 4.3.5. Inhibition of the MSC Immune Suppressor Capacities

MSCs are known to inhibit the anti-tumor immune response through their action on T- and B-lymphocyte activation and M2 macrophage polarization. Recent work showed that the CXCR1/2 blockade could prevent M2 macrophage polarization mediated by the CXCL1/2 ligands secreted by MSCs. The macrophages retained an M1 polarization, allowing tumor cell phagocytosis and tumor regression [184]. 

To conclude on MSCs, it remains difficult to establish a therapeutic strategy that specifically targets the CA-MSCs and abolishes their pro-tumor effects. However, numerous studies enabled the description of MSC biology, particularly in a tumor context, such as recruitment to the tumor sites, activations, and secreted molecules. Therefore, it is possible to consider in the short term the establishment of new therapies that would target signaling pathways activated in cancer cells and stromal cells by CA-MSCs. We can also decide to not eliminate MSCs but to use them as therapeutic agents. MSCs present well-established tumor-homing properties. They could, therefore, serve as promising systemic delivery tools. This idea was first suggested by Studeny et al. who showed that injected genetically engineered MSC cells (MSC–IFN-β) suppress the growth of pulmonary metastases, through the local production of IFN-β in the tumor microenvironment [190]. Since 2004, many researchers developed other genetically engineered MSCs producing TRAIL, IL-12, HSV-Tk, or sodium iodide symporter (NIS) and even a hepatocellular carcinoma-specific oncolytic adenovirus [191]. They both showed anti-tumor effects following injection of the genetically engineered MSCs to tumor-bearing animals. Taken together, all these studies revealed that, despite the fact that MSCs could promote tumor growth, they can serve as vehicles to specifically target anti-tumor molecules to tumor sites. This led to the proposal of clinical trials. Two trials are ongoing for ovarian cancer patients. The first involves the injection of genetically engineered MSCs expressing IFNβ, and the other involves oncolytic virus-loaded MSCs [192]. Finally, exosomes participate in tumor cell-to-MSC communication and are implicated in cancer pathogenesis. MSC exosomes could even display anti-tumor activities. They can act alone or as valuable vehicles for drug delivery. Exosomes derived from MSC enhance radiation effects observed in the control of metastatic spread of melanoma cells [193]. MSC-derived exosomes were shown to suppress oral cancer cell proliferation, invasion, and migration through microRNA-101-3p-targeting [194]. Lastly, in vivo taxol-loaded MSC-derived exosomes not only reduce the subcutaneous primary tumor but also prevent metastasis [195].

Therefore, if we cannot specifically circumvent CA-MSC pro-tumor effects, they can be used to target tumors.

## 5. CAFs

### 5.1. Overview on Normal Fibroblasts and Their Physiological Functions

Under physiological conditions, fibroblasts are encased within the interstitial extracellular matrix, representing the main component of the connective stromal tissue. Amongst stromal cells, fibroblasts represent the major cell type and contribute to the main functions of the connective tissue. Indeed, by producing extracellular matrix (ECM) proteins and ensuring their reorganization, as well as secreting several cytokines, chemokines, growth factors, and metabolites, they shape the tissue architecture while providing mechanical and metabolic support to the other cell types, and particularly epithelia. Fibroblasts also contribute to the regulation of the tissue immune response by regulating the recruitment of immune cells and sensitizing them to bacterial lipopolysaccharide. Thus, fibroblasts play crucial roles in tissue development, homeostasis, and repair [196]. 

Tissue-resident fibroblasts remain quiescent and display low proliferative and secretive rates. To date, the molecular signature of this quiescent state remains largely incomplete due to the lack of specific markers. Fibroblast-specific protein 1 (FSP1), also named S100A4, was routinely used until it was shown to also be expressed by macrophages and cancer cells [197]. Interestingly, a definite interplay between fibroblasts and their tissue of residency was highlighted [198]. Indeed, depending on their organ and site of origin, fibroblasts display differential transcriptional programs under the control of specific epigenetic alterations, demonstrating the tissue-specific plasticity of these cells and explaining the ECM particularities depending on the organs [199]. The ECM is mainly composed of collagens (type I, III, and V), elastin, fibrin, fibronectins, proteoglycans, and glycosaminoglycans, creating a three-dimensional network sheltering other stromal cells (immune, vascular, and glial cells, and neurons) [200]. In fact, ECM production, degradation, and interactions are most certainly the main fibroblast functions, making them essential players in tissue homeostasis regulation and remodeling. Therefore, as described for fibroblast types, ECM composition and organization also varies depending on the organ and localization within the tissue.

However, in response to injury, leading to wound healing, or under pathological conditions such as chronic inflammation, epithelial and immune cells secrete several cytokines (IFN-γ, TNF-α, IL-1, IL-6, and IL-8) and growth factors (TGF-β, PDGF (platelet-derived growth factor), and FGF2) activating fibroblasts, which changes their morphology from an elongated and fusiform shape to a wide cruciform shape, mainly due to the expression of αSMA (smooth muscle actin). These now-called myofibroblasts display increased migratory, secretive, and proliferative capacities but are, in turn, also more susceptible to epigenetic alterations [201]. In wounds, these activated fibroblasts actively remodel the ECM via the secretion of several ECM components and matrix-degrading proteases, including metalloproteinases (MMPs), cathepsins, urokinase–plasminogen system proteins, tissue inhibitors of MMPs (TIMPs), and aggrecanases, as well as protease inhibitors, facilitating the repair process [202]. In a physiological context, such as wound healing, fibroblast activation is reversible. Indeed, in response to other cell signals, they also express an intrinsic program aiming to control ECM remodeling and prevent any excessive scarring process, which could lead to fibrosis and dysfunctional regeneration, harmful for the tissue and the organ [203]. Epigenetic alterations, such as *RASAL1* promoter hypermethylation leading to Ras-GTP activity in renal fibrosis, seems to contribute to the mechanism underlying excessive fibroblast activation [204]. Aging and fibroblast senescence are also known to participate in tissue-resident fibroblast activation [205], and may play a role in malignancy.

### 5.2. Cancer-Associated Fibroblasts

CAFs are a complex of dynamically heterogeneous population of mesodermal cells in solid tumors, and represent a dominant cell type. CAF functions are different from those of tissue-resident fibroblasts, since these fibroblastic populations, with different origins, irreversibly develop an activated phenotype following chronic tumor-derived stimulation. This conversion stage is most likely epigenetic, as reported for skin fibroblasts upon leukemia-inhibitory factor chronic autocrine stimulation [206]. Intriguingly, global hypomethylation of the CAF genome was also reported, possibly involved in the upregulation of genes encoding for pro-tumorigenic secreted proteins [207]. Hence, CAFs are recognized as the main producers within the tumor of cytokines, chemokines, metabolites, and extracellular matrix-modifying enzymes such as matrix metalloproteases, which fuel tumor progression (growth and metastasis). Nevertheless, discrepancies regarding the pro- or anti-tumoral role of CAFs emerged and could be explained by either their different origins or their dynamic phenotypic evolution during tumor progression, implying heterogeneity in markers and in functions.

CAFs present in the tumor bed are a mix of mesodermal cells, mainly originating from the activation of tissue-resident quiescent fibroblasts, which expand in the host tissue upon chronic stress induced by the developing tumor. CAFs may also originate from distant sources such as the bone marrow, thereby forming a functionally distinct stromal cell population as reported in breast cancer [208]. Trans-differentiation of local stromal cells was also described to give rise to CAF-like populations, through program switches called endothelial-to-mesenchymal transition for pericytes or endothelial cells [207], and epithelial-to-mesenchymal transition (EMT) for epithelial cells [207]. Since CAFs display normal allelotypes, at least in pancreatic cancer [209], EMT-derived CAFs may not arise from tumor cells directly but from normal epithelial cells, which receive paracrine signals that induce alterations in their epithelial programs from the adjacent tumor, (i.e., cell polarity, cell-to-cell and cell-to-ECM junctions). Alongside the generation of CAFs, this EMT program may dampen the so-called epithelial intercellular surveillance, thereby fostering tumor progression [210]. Nevertheless, the biological origin of the different hypothetic CAF populations is still under debate due to the lack of specific markers for each population. Transcriptional analyses at the single-cell level in tumors will soon lead to the identification of molecular signatures of the different stromal cell populations and specific markers, which are necessary to clearly identify their biological origins (for example, through genetic lineage tracing) and respective functions. 

Numerous non-specific markers for CAFs were reported due to the different cell origins. Tissue-resident fibroblasts exhibit different organ-specific transcriptomic profiles [198], implying organ-specific markers for CAFs derived in the respective tumors. For example, αSMA^+^ CAFs are predominant in pancreatic cancer arising from resident fibroblastic stellate cell activation [211], whereas PDGFRα^+^ CAFs reflect in melanoma the activation and expansion from dermal fibroblasts [212]. Lack of specific markers for each hypothetical CAF population implies the overlap of different markers, such as FSP1/S100A4 and αSMA markers in pancreatic or breast CAFs [207]. Based on the relative co-expression of six fibroblastic markers (αSMA, FAPα (fibroblastic-activated protein α), integrin β1/CD29, S100-A4/FSP1, PDGFRβ, and CAV1 (caveolin 1), Costa et al. identified, by fluorescent analysis cell-sorting, four distinct CAF subsets in human breast cancer that accumulate differentially in tumors and juxta-tumors, one of these being functionally endowed with immunosupressive properties [213]. 

Neuzillet et al. proposed the “pCAFassigner” classification to segregate pancreatic patient tumors into prognosis groups depending on expression levels of three different markers, i.e., periostin, MYH11 (myosin-11), and podoplanin [214]. Prognostic signatures of stroma based on the identification of different marker sets (gene expression studies) were uncovered in patients to predict survival, recurrence, or resistance to neoadjuvant chemotherapies, in several cancers; however, the lack of uniformity between tumors, probably reflecting CAF heterogeneity, will require the design of specific signatures for each pathology [215]. 

Markers most commonly used to identify CAFs relate to the main biological functions of CAFs, as the main producers of ECM components (collagens I and II, fibronectin, tenascin-C, and periostin), remodeling enzymes (lysyl oxidases, matrix metalloproteases, and their inhibitors), secreted growth factors (e.g., TGFβ, VEGF, PDGFs, EGF, FGFs, CTGF (connective growth factor), WNTs), and cytokines (IL-6, CXCL12), and as contractile cells (cytoskeletal components such as αSMA). Intriguingly, distinct spatial tissue distributions in pancreatic cancer between different reported CAF populations expressing specific markers may result from different dialogues with tumor cells; inflammatory IL-6^+^ CAFs, distant from tumor cells, receive an IL-1 signal which triggers the JAK–STAT pathway activation and promotes a CAF inflammatory state, whereas CAFs residing in the vicinity of tumor glands mainly receive a TGFβ signal that favors a myofibroblastic SMA^+^ state by antagonizing the IL-1 signal [216,217]. This example provides a mechanism through which distinct spatial fibroblast niches are established in the tumor microenvironment, and also illuminates the importance of studying CAF functional heterogeneity by considering their spatial distribution [218]. Very recently, single-cell RNA sequencing to pancreatic tumor tissue identified distinct CAF subsets, confirming the existence of distinct myofibroblastic and inflammatory CAF subsets, and providing novel marker genes for these cells (TNC, TGFB1, THY1, TAGLN, COL12A1, and PDGFRB for myofibroblastic CAFs, or CLEC3B, COL14A1, GSN, LY6C1, and CXCL12 for inflammatory CAFs). This study also identified a new CAF subtype that expresses MHC class II-related genes and specific markers (SLPI, SAA3, CD74, H2-AB1, NKAIN4, IRF5), and induces T-cell receptor ligation in CD4^+^ T-cells in an antigen-dependent manner [219]. Since these CAFs lack the co-stimulatory molecules required to induce T-cell proliferation, they are hypothesized to inhibit optimal T-cell response. 

Pro- and anti-tumoral functions were attributed to CAFs in cancers, most certainly due to their different origins, and phenotypic evolution during tumor progression. The two-step activation model differentiating normal active fibroblasts (part I) from CAFs further supports the concept of the fibroblast’s dual behavior against or supporting tumor cell growth during cancer initiation or progression, respectively. If one considers tumors as a wound that never heals, it is conceivable that acutely (reversibly) activated fibroblasts will restrain tumor initiation (such as in wound-healing process), whereas chronically (irreversibly) activated fibroblasts will sustain tumor progression [220].

Studies describing pro-tumorigenic functions for CAFs are more numerous than those attributing restraining activities. This may result from CAF heterogeneity within a same tumor and from ad-mixing experiments using ex vivo primary CAF cultures established from patient tumors; a possible enrichment in these cultures of tumor-promoting CAFs (at the expense of tumor-restraining CAFs) may result from the secretion by these CAFs of pro-survival factors that promote not only tumor cell growth and invasion but also autocrine growth stimulating loops in CAFs [221]. 

CAF pro-tumorigenic functions are mainly triggered by their altered secretomes, whereby paracrine signaling from CAFs boosts tumor progression by positively impacting on cancer cell survival, proliferation, stemness, metabolism, metastasis-initiating capacity, and resistance to therapies (reviewed in Reference [221]). Soluble factors reported to be involved in such dialogues largely comprise soluble (growth factors, cytokines, and chemokines) and insoluble proteins (extracellular matrix (ECM)), such as CXCL12, TGFβ, FGFs, HGF, IL-6, or periostin [222,223,224,225,226,227]. These proteins act on tumor cells, but also indirectly stimulate tumor growth by impacting other stromal cells, such as endothelial cells, thereby remodeling the tumor vasculature, or immune cells, favoring the specific recruitment or intra-tumor polarization of pro-tumorigenic inflammatory cells (reviewed in Reference [228]). An explanation for altered protein secretomes in pancreatic CAFs was reported to depend on exacerbated mTOR-induced protein synthesis, the targeting of which revealed a therapeutic benefit in vivo, restraining tumor growth and metastasis, as well as re-sensitizing cancer cells to chemotherapy [229,230]. Other paracrine factors secreted by CAFs that support tumor growth by improving tumor energy synthesis include lipids, such as recently demonstrated in pancreatic tumors, where CAFs undergo a dramatic lipid metabolic shift [231], or metabolites such as lactate or ketone bodies [232]. 

ECM proteins and ECM-remodeling enzymes secreted by CAFs not only support cancer cell survival by stimulating integrin receptor signaling in cancer cells or by releasing ECM-sequestered growth factors, respectively (reviewed in Reference [233]), but compose a crosslinked collagen fiber network organized into a parallel orientation forming stiff “migration highways” that promote the directionality and velocity of cancer cells (reviewed in Reference [234]), through cooperative collective invasion with CAFs [235]. Communication between CAFs and cancer cells also involve extracellular microvesicles that transmit biological information by carrying microRNAs, long non-coding RNAs (lncRNAs), proteins, or metabolites, both locally and at distance, thereby involved in preparing distant pre-metastatic niches (reviewed in Reference [236]. Such cargo functions for extracellular microvesicles may in the future have clinical utility.

Whether CAF populations involved in restraining tumor progression correspond to fibroblasts resistant to irreversible conversion into CAFs, or to distinct CAF subsets remains unknown. There is a growing body of evidence suggesting that cancer cells influence normal fibroblasts to suppress their tumor-suppressing activity. In a normal setting, stromal cells secrete TGF-β inhibitors [237] or the β-catenin-signaling inhibiting factor SLIT2 [238], or express safeguard checkpoints (e.g., p53) [239], which, upon epithelial stress (e.g., carcinogenesis initiation), will restrain epithelial cell proliferation. Inactivation of these stromal checkpoints was correlated with tumor progression. Tumor-suppressing functions of CAFs were reported in pancreatic cancer, in which targeting the believed pro-tumor hedgehog signaling pathway in CAFs, or genetic depletion of αSMA^+^ CAFs from tumors unexpectedly enhanced tumor progression. These results, therefore, suggest that Hedgehog-responsive and myofibroblastic αSMA^+^ CAFs may comprise two tumor-suppressive subsets. Interestingly, these two subsets may belong to the same CAF population since it was shown, using a Hedgehog pathway reporter allele, that nearly all αSMA^+^ fibroblasts are Hedgehog-responsive. Further supporting the tumor restraining effect of the Hedgehog pathway, a clinical trial that combined chemotherapy with IPI-926, a small-molecule inhibitor of the Hedgehog pathway, was aborted due to a shortened patient overall survival (NCT01130142) [240]. 

### 5.3. CAFs Targeting

Therapeutically targeting CAFs in cancer may have its use, as they are (i) the most numerous cell type in many fibrotic aggressive tumors, e.g., pancreatic cancer, (ii) endowed with tumor-growth features, and (iii) genetically stable and consequently less likely than tumor cells to acquire therapeutic resistance. Nevertheless, until CAF tumor-suppressive subsets are clearly identified and characterized, such targeting strategies may not be as promising as expected.

#### 5.3.1. Depletion

Strategies attempting to ablate CAFs from tumors using cell subset markers provided promising therapeutic benefits when targeting the FAPα^+^ fibroblasts in pancreatic carcinoma [241], but not the αSMA^+^ subsets [242], further strengthening the functional diversity of these distinct CAF populations. Preclinical studies targeting FAPα^+^ fibroblasts with either a DNA vaccine against FAPα, an FAPα enzymatic inhibitor, an FAPα antibody, or chimeric antigen receptor T-cells targeting FAPα, also provided promising preclinical results in mice [243], but failed in clinical trials (phase II in metastatic colorectal cancer using a humanized anti-FAPα antibody or the inhibitor PT-100) [243,244]. Outcomes of a novel strategy targeting FAPα^+^ cells using a recombinant fusion protein that targets an engineered variant form of IL-2 to human FAPα^+^ cells, to stimulate a local immune response, are awaited (phase I and II clinical trials in combination with EGFR inhibitor (NCT02627274) or anti-PD-L1 antibody (NCT03386721)). Another CAF subset (CD10^+^ GPR77^+^) was identified in chemotherapy-resistant breast tumors, the targeting of which using a neutralizing anti-GPR77 antibody abolished tumor formation and restored tumor chemosensitivity of patient-derived xenografts [245]. CAFs from pancreatic cancer were described to express the somatostatin receptor sst1, whose activation using an sst1 agonist (SOM230, pasireotide) inhibited the mTOR protein synthesis pathway in CAFs, thereby dramatically decreasing the secretion of pro-tumor secreted proteins, including IL-6, inhibiting tumor growth and metastasis, and restoring chemosensitivity in preclinical mouse models including PDX [229,230]. Clinical phase I trials using pasireotide with the chemotherapies, gemcitabine or FOLFIRI, showed manageable safety profiles in patients with gastrointestinal malignancies and provided preliminary signals of activity. Larger phase II trials are, therefore, warranted. This therapeutic approach using SOM230 targeting CAF protein synthesis of secreted factors sounds wiser than drugs targeting single secreted proteins, since compensatory functions between the large panel of CAF secreted factors may occur. An example of such a strategy targets the HGF-cMet signal using a neutralizing anti-HGF antibody or cMet inhibitor, demonstrating early signs of anti-tumor activity in a phase I clinical trial [246,247].

#### 5.3.2. Reprogramming

Strategies to reprogram CAFs to re-normalize tumor stroma targeted the epigenetic machinery, or pathways involved in CAF activation, e.g., TGF-β-mediated Smad2/3 signaling, using the viTAMsin-D derivative or the bioactive lipid lipoxin A4 [248,249], providing encouraging growth suppression in preclinical models. 

Strategies using CAF’s enzymatic features as a means of drug delivery are being developed, such as a prodrug or micelles encapsulating a chemotherapeutic drug, designed to be cleaved using the protease activity of CAF-expressed FAPα or MMPs, respectively [250,251]. LRRC15^+^ (leucine-rich-repeat-containing) CAFs were targeted using an LRRC15–antibody drug conjugate (phase I ABBV-085) to deliver an anti-mitotic drug. Recent approaches which may delay peritoneal metastasis were designed to use CAFs as ecological traps that attract disseminated cancer cells, by encapsulating live CAFs into microparticles that are intraperitoneally injected and then retrieved using the microparticle’s magnetic properties [252].

## 6. Conclusions

In this review, we highlighted cellular components of the TME, notably TAMs, T-regs, MSCs, and CAFs, which can be pro-tumorigenic by promoting cancer cell survival, immune system evasion, chemoresistance, migration, and metastasis. Interestingly, these infiltrated cells share the ability to produce the same cytokines involved in all these pro-tumorigenic mechanisms, including, for instance, TGF-β, one of the key mechanisms suppressing immune effectors including NK cells [253]. Therapeutically targeting these cells or the soluble factors they secrete may have great benefits in cancer. The most promising preliminary results were obtained by exhausting TAMs with anti-CSF-1R or abrogating T-reg functions with anti-PD-1. Even if no real response was achieved in vivo with monotherapy, combination strategies, in particular anti-CSF-1R/chemotherapy, showed a good tumor regression in patients. Targeting CAFs or MSCs could also be an interesting breakthrough. However, strategies available to date are not as promising as expected for targeting CAFs and not specific enough for targeting MSCs. Development of combination therapies acting both to suppress TME barriers and stimulate immune responses remains the most promising for patient survival. Many advances are yet to come with a better understanding of tumor immunology, identification of new targets, and optimization of immunotherapy protocols. 

## Figures and Tables

**Figure 1 ijms-20-04719-f001:**
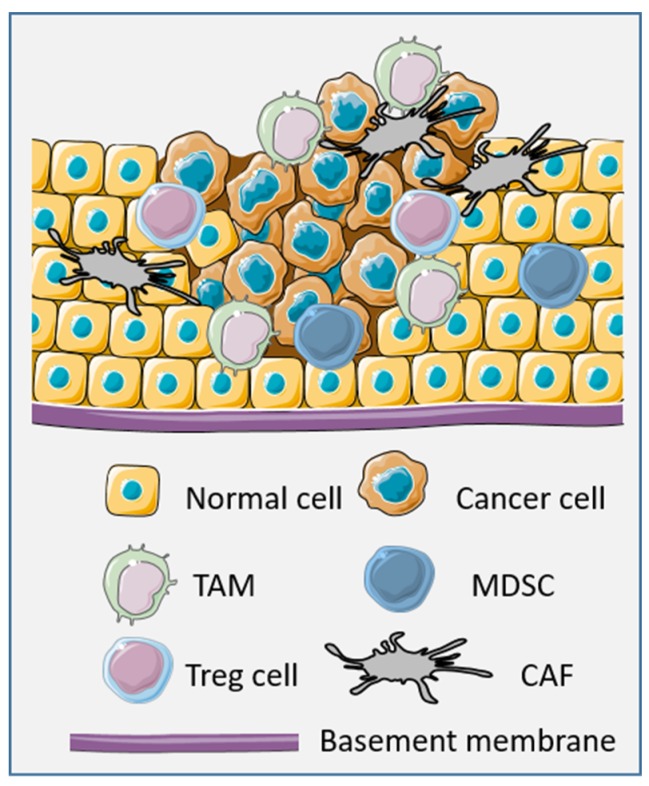
Representation of the tumor microenvironment (TME) with tumor-associated macrophages (TAMs), mesenchymal stromal/stem cells (MSCs), regulatory T-cells (T-regs), and cancer-associated fibroblasts (CAFs) infiltrating the tumor.

**Figure 2 ijms-20-04719-f002:**
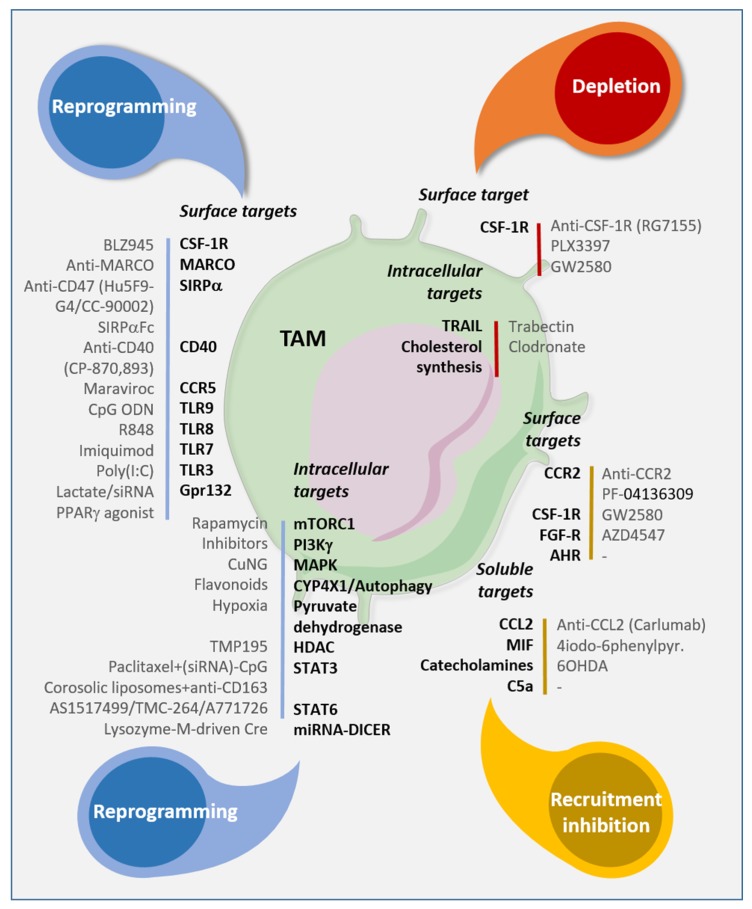
Targeting strategies to reprogram, eliminate, and inhibit TAM recruitment. Antibodies or molecules available to target surface, intracellular or soluble molecules involved in the phenotype, functions, and recruitment of TAMs in the TME (as outlined in the text).

**Figure 3 ijms-20-04719-f003:**
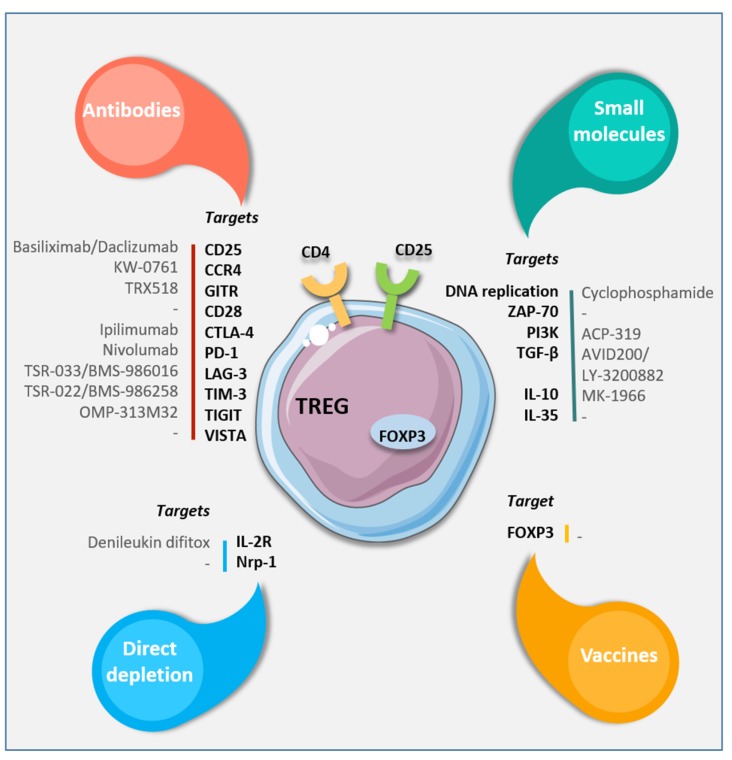
Targeting strategies to eliminate or modulate T-reg functions. Available antibodies, small molecules, or vaccines specific for different cell surface or intracellular targets (as outlined in the text).

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
