# Peer review of "Latest Advances in Targeting the Tumor Microenvironment for Tumor Suppression"

_ijms, 2019, doi:10.3390/ijms20194719_

Round 1

Reviewer 1 Report

Laplagne C. and colleague provided an exhaustive and well written overview on latest discoveries aimed to target tumor microenvironment, especially the cell compartment represented by TAM, Treg, MSC and CAF. Interestingly as described by the authors, these cells share TGF-b production, one of the key mechanism suppressing anticancer immune effectors including NK cells. Since the authors cited this TGF-beta-mediated inhibitory mechanism against NK cells, some related references are missing and might be added. All the comments are listed below.

References to be added

or references regarding possible impact of TGF-b on NK cell function and differentiation:

SMAD4 impedes the conversion of NK cells into ILC1-like cells by curtailing non canonical TGF-b signaling

Cortez V.S. et al. Nat. Immunol. 2017

Tumor immunoevasion by the conversion of effector NK cells into type 1 innate lymphoid cells

Gao Y. et al. Nat Immunol. 2017

TGFβR1 Blockade with Galunisertib (LY2157299) Enhances Anti-Neuroblastoma Activity of the Anti-GD2 Antibody Dinutuximab (ch14.18) with Natural Killer Cells.

Tran HC et al., Clin Cancer Res 2017

Lane 120: it could be more appropriate “Differentiation and Depletion” rather than “Depletion” alone.

Lane 172-174 the sentence is misleading “...become anti-tumor and…” rather than “…be anti-tumor and…”

Some typing errors

Please explain acronym only the first time you use it

Author Response

I would like to thanks Reviewer 1 for his constructive remarks.

I added the 3 recommended references concerning the TGF-b and the NK cells in the conclusion. (blue highlight)

I corrected the lines 120 and 172. (blue highlight). Please see the attachment.

I removed the multiple acronym explanations and we tried to review the typing errors.

Reviewer 2 Report

RE: Latest advances in tumor microenvironment targeting for tumor suppression

This a long comprehensive review.

However many citations are missing -for example the first whole paragraph of the introduction has only 3 citations.

Also incorrect statements see attached pdf

Rewrite abstract, and last paragraph - make clearer

Line 44-45 “ In the last case, infiltrated T-cells involved in immune system regulatory pathways are called regulatory T-cells (T-regs), doesn’t make sense T-regs are a subset known to be immunosuppressive, pro-tumor with certain markers including CD4, FOXP3 and CD25. In the next sentence tumor associated macrophages can be both pro or anti-tumour. Needs reworking with the right citations.

The language used in manuscript is not scientific and needs to be revised

Line 54 “high range of phenotypes’ rather something like site specific phenotypes depending on the tissue, i.e. ..microglia in the ..

Line 64 “Being involving”

Line 81 “Actually”

Line 90 “at the beginning of the tumor” –to early in tumor initiation

Needs to be rewritten and appropriate citations added in

Add tables in to summerise text maybe a table of clinical trials, drug and tumour type and target would be useful

Add paragraphs pg 10

Author Response

I would like to thanks Reviewer 2 for constructive remarks. I tried to modify the article according to his suggestions. 

All modifications are highlighted in green, please see the attachment.

I added references as asked by Reviewer 2.

However, I didn't add a table of all clinical trials, which would be a huge picture regarding the 4 cellular components discussed in this review. Moreover, I discussed in this review several new advances with new targets which are not associated with clinical trials.